# 3D Medical Axial Transformer: A Lightweight Transformer Model for 3D Brain Tumor Segmentation

**Cheng Liu**[1]                                    9645269447@edu.k.u-tokyo.ac.jp

**Hisanori Kiryu**[1]                                    kiryu-h@k.u-tokyo.ac.jp

[1] *Department of computational biology and medical sciences, The University of Tokyo, Tokyo, Japan*

**Editors:** Accepted for publication at MIDL 2023

## Abstract

In recent years, Transformer-based models have gained attention in the field of medical image segmentation, with research exploring ways to integrate them with established architectures such as Unet. However, the high computational demands of these models have led most current approaches to focus on segmenting 2D slices of MRI or CT images, which can limit the ability of the model to learn semantic information in the depth axis and result in output with uneven edges. Additionally, the small size of medical image datasets, particularly those for brain tumor segmentation, poses a challenge for training transformer models. To address these issues, we propose 3D Medical Axial Transformer (MAT), a lightweight, end-to-end model for 3D brain tumor segmentation that employs an axial attention mechanism to reduce computational demands and -distillation to improve performance on small datasets. Results indicate that our approach, which has fewer parameters and a simpler structure than other models, achieves superior performance and produces clearer output boundaries, making it more suitable for clinical applications. The code is available at https://github.com/AsukaDaisuki/MAT.

**Keywords:** Deep learning, 3D brain tumor segmentation, 3D Transformer, axial attention, self-distillation

## 1. Introduction

Medical image segmentation is a key component in computer-aided diagnosis and a fundamental procedure in medical image processing (Doi, 2007). It helps clinicians make more accurate diagnoses and treatment decisions by segmenting organs or tumors in medical scans. With the development of convolutional neural networks (CNNs), Unet (Ronneberger et al., 2015) emerged as a popular medical image segmentation network with its simple U-shaped structure and innovative skip connections design. Many variations of Unet have been developed, including V-Net (Milletari et al., 2016), Res-Unet (Zhang et al., 2018), H-Dense-Unet (Li et al., 2018) and 3D-Unet (Çiçek et al., 2016) for 3D medical image segmentation. Specifically, the nnUnet (Isensee et al., 2021) model has demonstrated state-of-the-art performance in a wide range of tasks, including brain tumor segmentation. CNNs have achieved success in medical image segmentation, but struggle with long-range dependencies and processing global context (Vaswani et al., 2017). This is especially problematic in brain tumor images, where doctors need to combine information from multiple regions for diagnosis (Valanarasu et al., 2021).

Transformer-based models, utilizing self-attention mechanisms, have become popular in natural language processing and set state-of-the-art benchmarks in recent years (Devlin

et al., 2018; Brown et al., 2020). These models are able to efficiently compute dependencies between sequential inputs, even when distant from each other, addressing the problem of long-range dependencies which traditional convolutional models struggle with. In computer vision, the Vision Transformer (ViT) (Dosovitskiy et al., 2020) adapted the Transformer for image classification tasks with successful results. Subsequently, Swin-Transformer (Liu et al., 2021) has combined the sliding window concept of CNNs with self-attention mechanisms to form the Transformer using shifted window methods, capable of handling downstream tasks such as classification, detection, and segmentation.

In the field of medical image segmentation, Trans-UNet (Chen et al., 2021) pioneered the use of transformer-based models by integrating them into the traditional Unet architecture. Another notable approach is Medical Transformer (MedT)(Valanarasu et al., 2021), which uses gated axial attention for efficient semantic feature learning. Additionally, Swin-Unet (Cao et al., 2021) leveraged the efficient structure of the Swin-Transformer to achieve superior performance. However, the high computational cost of Transformer models with 3D inputs has limited their application to mainly 2D slices. UNETR(Hatamizadeh et al., 2022b) and Swin-UNETR (Hatamizadeh et al., 2022a) are exception, achieving good results on 3D medical images, but at the cost of high GPU memory consumption. To mitigate this, CoTr (Xie et al., 2021) proposed dividing 3D images into smaller blocks, but this also sacrifices semantic information. AFTer-Unet (Yan et al., 2022) utilized several neighboring axial slices as a 3D input image, reducing resource overhead but limiting global dependencies and requiring extra pre-processing.

We propose the 3D Medical Axial Transformer (MAT) model for efficient 3D brain tumor segmentation. MAT builds upon the success of MedT by introducing three one-dimensional gated axial attention mechanisms within the Transformer module, which effectively reduces GPU resource consumption and memory cost. In contrast to AFTer-Unet and MedT, MAT focuses on the entire depth axis and uses an innovative self-distillation training scheme to improve performance on small datasets, while helps the axial attention learn the correct position and semantic information in all three dimensions. Our contributions include: (1) an end-to-end model that eliminates the need for pre-processing and allows for the direct use of 3D images, (2) a 3D self-attention module utilizing axial attention mechanism, and (3) the use of self-distillation with a warm-up schedule, a novel approach in medical image processing, which improves the ability of the transformer module to learn information from small-scale datasets. Particularly, our experiments reveal that by using the self-distillation method, the axial attention can effectively learn the correct information in all three axis, while designed warm-up schedule allows the model to converge quickly, leading to more promising results.Our experiments indicate that above approach is effective.

## 2. Methodology

The proposed 3D Medical Axial Transformer (MAT) model utilizes a Transformer architecture with axial attention mechanisms in the encoder component, in combination with CNNs. The encoder is connected to a decoder, composed of CNNs, through skip connections at various resolutions. It is noteworthy that, due to the constraints imposed by the 3D input on the mini-batch size, group normalization (GN) is employed as a regularization technique in MAT, as opposed to batch normalization. The overall architecture of the

model is depicted in Figure 1(a). In this section, we present a thorough description of the encoder and decoder of the proposed MAT model.

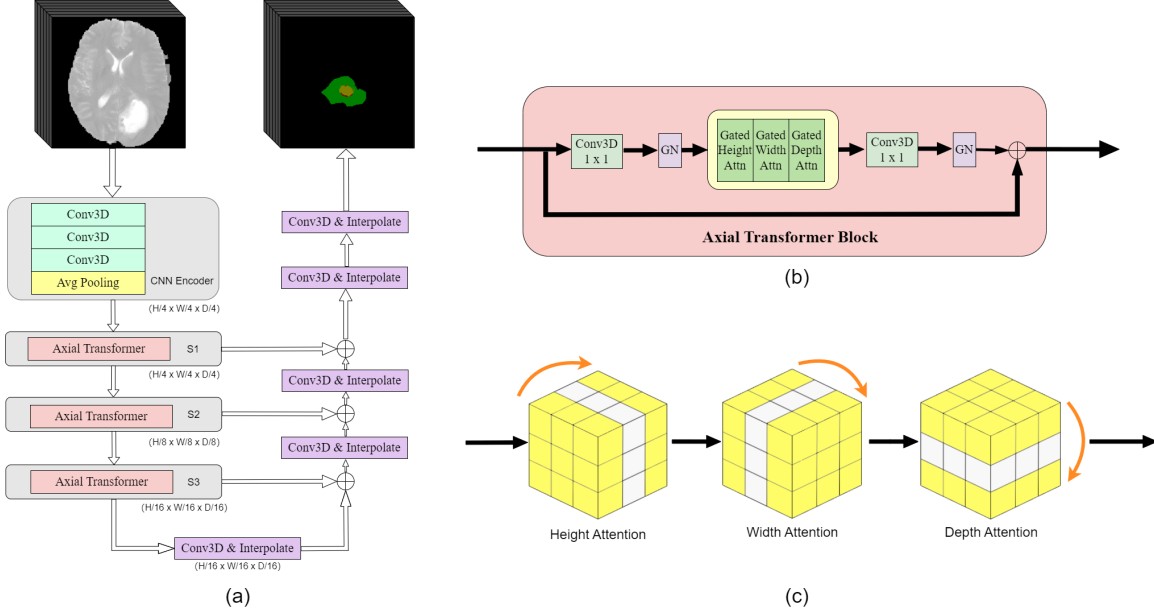

Figure 1: (a) Architecture overview of MAT (b) designs of Axial Transformer Block (c) Schematic diagram of the Axial Attention calculation method

## 2.1. CNN Encoder

MAT utilizes a minimal number of CNN blocks to extract features before passing the input to the Transformer encoder, following the design of MedT (Valanarasu et al., 2021). This approach allows the Transformer encoder to better learn semantic information and reduces the number of model parameters. The CNN encoder consists of one ($7{\times}7{\times}7$) convolutional block and two ($5{\times}5{\times}5$) convolutional blocks. The initial convolutional block ($7{\times}7{\times}7$) with a stride of 2 improves feature extraction, as established in (Simonyan and Zisserman, 2014), with a group normalization layer added between the convolutions. In contrast to MedT, an average pooling layer of size ($2{\times}2{\times}2$) with stride 2 is implemented between the CNN encoder and the Transformer encoder to extract further features while conserving memory resources.

Suppose the input to the encoder is a 3D image $x \in \mathbb{R}^{C \times D \times H \times W}$, where (D × H × W) is the image resolution and C is the channel of the input (e.g. C = 4 channels for MRI). After the CNN encoder, the output should be $x_{CNN} \in \mathbb{R}^{C^{cnn} \times \frac{D}{4} \times \frac{H}{4} \times \frac{W}{4}}$, where $C^{cnn}$ is the number of channels output from the last convolution layer.

## 2.2. Axial Transformer Encoder

After using CNNs to extract shallow image features, we introduce Transformer encoders to facilitate the learning of deep semantic information. To reduce computational complexity,

we implement the Transformer blocks using 3D axial attention, known as Axial Transformer, which allows for self-attention computation in all three dimensions, resulting in a comprehensive 3D modeling of the images, as illustrated in Figure 1(b). This design choice effectively reduces resource consumption while enabling the model to effectively process and understand the full context of the medical images.

### 2.2.1. Axial Attention

In this work, we propose the use of three axial attention mechanisms in MAT for efficient 3D self-attention computation. This approach decomposes the 3D attention calculation into three 1D calculations in the three dimensions, reducing computational complexity, as illustrated in Figure 1(c). We also incorporate a learnable positional bias term and relative positional encoding in the self-attention module to effectively capture positional information in the 3D medical images. This approach builds upon previous research (Ho et al., 2019; Wang et al., 2020) that have shown the effectiveness of axial attention in capturing semantic information in 3D images. As an example, the output of the height axial attention is illustrated as Eq (1), Where $r^q$, $r^k$, and $r^v$ are the relative position encoding for queries, keys, and values respectively.

$$y_{\text{height}_{ijk}} = \sum_{h=1}^{H} \text{Softmax}\left(q_{ijk}^T k_{ihk} + q_{ijk}^T r_{ihk}^q + k_{ihk}^T r_{ihk}^k\right)(v_{ihk} + r_{ihk}^v) \tag{1}$$

To address the challenge of training the Transformer on small datasets and ensuring adequate position encoding, we incorporate a gating mechanism, as proposed in previous research (Valanarasu et al., 2021). The gated axial attention on the height axis is as follows.

$$y_{\text{height}_{ijk}} = \sum_{h=1}^{H} \text{Softmax}\left(q_{ijk}^T k_{ihk} + G_q q_{ijk}^T r_{ihk}^q + G_k k_{ihk}^T r_{ihk}^k\right)\left(G_v^1 v_{ihk} + G_v^2 r_{ihk}^v\right) \tag{2}$$

Where $G_q$, $G_k$, $G_v^1$, and $G_v^2$ are gates for queries, keys, and values, respectively. These gates act as learnable parameters that effectively control the final output of attention. In general, the gating mechanism limits the output of poor position encoding and gives higher weights to those that have been learned relatively well (Valanarasu et al., 2021).

### 2.2.2. Architecture

Our Axial Transformer block is based on the traditional Transformer design and features 3D convolutional layers, group normalization, and axial attention for height, width, and depth, as depicted in Figure 1(b). The output of the last GN layer is connected to the input via a skip connection (Vaswani et al., 2017; He et al., 2016).

Medical images have been found to require higher accuracy rather than a complex method for processing, based on research of various datasets (Isensee et al., 2021). To address this, we have chosen to use a lightweight model with fewer blocks to avoid issues such as overfitting or difficulty in practical application. We use 3 modules for axial attention computation and divide it into three stages: S1, S2, and S3, with an average pooling layer (strides = 2) behind each of the modules to reduce feature map resolution, except for T1 to prevent loss of important semantic information at an early stage.

The input is assumed to be $x_{CNN} \in \mathbb{R}^{C^{cnn} \times \frac{D}{4} \times \frac{H}{4} \times \frac{W}{4}}$ (the output of the CNN encoder). Each stage expands the number of channels by a factor of two through multi-headed attention, so the output of each stage turns to be $\left(2C^{cnn} \times \frac{D}{4} \times \frac{H}{4} \times \frac{W}{4}\right)$, $\left(4C^{cnn} \times \frac{D}{8} \times \frac{H}{8} \times \frac{W}{8}\right)$ and $\left(8C^{cnn} \times \frac{D}{16} \times \frac{H}{16} \times \frac{W}{16}\right)$.

## 2.3. CNN Decoder

Our CNN decoder is designed using the Unet architecture (Ronneberger et al., 2015), with a structure largely symmetric to the Axial Transformer encoder. It comprises five CNN-interpolation blocks, where the first block has, a stride of 1, while the remaining blocks have a stride of 2. The ReLU activation function is employed, and the image resolution is gradually increased by a factor of two with each block on the last four blocks. A skip connection is utilized to connect the encoder and decoder to restore details lost during downsampling. Instead of the traditional four skip connections in the up-sampling blocks, we used only three, discarding the topmost one as suggested by (Guo et al., 2022) to avoid negative impact from noisy shallow layer semantic information.

Let the final output of Axial Transformer encoder be $X_{encoder}$ with a resolution of $\left(\frac{D}{16} \times \frac{H}{16} \times \frac{W}{16}\right)$, then the output after five CNN-interpolation blocks will be $\left(\frac{D}{16} \times \frac{H}{16} \times \frac{W}{16}\right)$, $\left(\frac{D}{8} \times \frac{H}{8} \times \frac{W}{8}\right)$, $\left(\frac{D}{4} \times \frac{H}{4} \times \frac{W}{4}\right)$, $\left(\frac{D}{2} \times \frac{H}{2} \times \frac{W}{2}\right)$ and $(D \times H \times W)$, which means that the resolution reduction is completed. At the last, a convolutional layer is used for classification, thus completing the downstream task of image segmentation.

## 2.4. Self-distillation for Regularization with warm-upped DLB

To address the challenge of training Transformer-based models on small datasets, we propose the use of self-distillation, a technique that utilizes the model's own output as a soft target for learning (Zhang et al., 2019). This approach allows for the optimization of the model directly through the training schedule, without the need for extensive modification of the architecture or the use of a large teacher model like traditional knowledge distillation (Hinton et al., 2015). (Bhat et al., 2021; Gani et al., 2022) has shown that self-distillation can be effective in improving performance on small datasets, and may also act as a regularization-like effect to aid in model training(He et al., 2022). Recently, (Shen et al., 2022) proposed a new method of implementing self-distillation, named DLB, where the model from the previous iteration is used as the teacher for the current iteration. The loss function for this method is formulated as:

$$\mathcal{L}_{LB} = \frac{1}{n} \sum_{i=1}^{n} \mathcal{T}^2 \cdot D_{KL}\left(p_i^{\mathcal{T},t-1} \| p_i^{\mathcal{T},t}\right) \tag{3}$$

Where n is the number of samples in a batch, $\mathcal{T}$ is the temperature of the distillation, and $p_i$ is the predicted distribution of each sample (Shen et al., 2022).

To date, there is a lack of literature on the application of self-distillation to the task of medical image segmentation. We propose to integrate the DLB loss as an additional term to the primary loss function in our model architecture, which is a combination of weighted cross-entropy loss and Dice loss, commonly used in medical image segmentation tasks. The

overall loss function for our model is presented in Eq (4).

$$\mathcal{L}_{MAT} = 0.4 \cdot \mathcal{L}_{CE} + 0.6 \cdot \mathcal{L}_{\text{Dice}} + \alpha \cdot \mathcal{L}_{LB} \tag{4}$$

The implementation of our DLB follows the approach of the original DLB, with a constraint of maintaining half of the mini-batch consistency between consecutive iterations. However,due to the Transformer models tending to converge more slowly and being more susceptible to instability during the early stages of training, the use of DLB may instead lead to the problem of each iteration affecting each other and eventually being difficult to converge. To mitigate these issues, we implement a dynamic warm-up schedule for the distillation coefficient $\alpha$ and temperature $\mathcal{T}$. Specifically, the value of $\alpha$ is set to 0 for the first 50 training epochs, before being set to 1. In the following epoch, $\mathcal{T}$ is linearly increased from 1 to 2, with the aim of placing more emphasis on the distribution of negative samples during later stages of training.

## 3. Experiments

### 3.1. Setup

#### 3.1.1. Dataset

The BraTS datasets from the MICCAI Brain Tumor Segmentation Challenge contain multimodal 3D brain MRI scans annotated with ground truth segmentations of tumor regions by physicians. The datasets consist of four MRI modalities per case (T1, T1ce, T2, and FLAIR), with annotations of four tumor subregions consolidated into three sub-regions: whole tumor (WT), tumor core (TC), and enhancing tumor (ET). BraTS2018 (Menze et al., 2014a; Bakas et al., 2017c, 2018) was collected from 19 institutions and includes both low-grade and high-grade gliomas. The study focuses on high-grade gliomas, as it provides an opportunity to demonstrate the advantages of the model on small datasets. The HGG group contains 210 samples randomly split into 180 training cases and 30 testing cases. In comparison, BraTS2021 (Bakas et al., 2017c; Baid et al., 2021; Menze et al., 2014b; Bakas et al., 2017a,b) is larger and randomly split into 1200 training cases and 200 testing cases to enable a comparison of our method's performance with larger datasets.

#### 3.1.2. Evaluation Metrics

As in previous studies (Valanarasu et al., 2021; Yan et al., 2022), we also used the Dice score and the 95% Hausdorff Distance to assess the overall accuracy of segmentation as well as the surface accuracy. The formulas of Dice and HD95 are defined in appendix section .

#### 3.1.3. Implementation details

The images in the BraTS 2018 and 2021 datasets were re-sized to $(160{\times}224{\times}224)$ and $(128{\times}160{\times}160)$ for consistency and to fill empty slices. Data augmentation, including random flips and rotations, was applied with a 50% probability to enhance the model's fitting ability. The hyperparameters of the model were tuned via 5-fold cross-validation on the training set, and then applied to train the full training set, yielding the best model. The AdamW optimizer (Loshchilov and Hutter, 2018) with a weight decay of 10-5 was utilized

with a warm-up schedule for the learning rate (Gotmare et al., 2018), where the learning rate grows linearly from 0 to 10-3 for the first 10 epochs, followed by cosine anneal (Loshchilov and Hutter, 2016) to complete the learning rate decay. In the Transformer encoder, we employed 1, 2, and 4 blocks for stage S1, S2, and S3, respectively. For the sake of fairness, in our experiments, all models were trained for an equal number of 250 epochs.

## 3.2. Results on BraTS2018

Table 1: Dice scores and HD95 of different methods on the BraTS2018 dataset (testing).

| Metrics | Params | WT | | TC | | ET | | Mean | |
|---|---|---|---|---|---|---|---|---|---|
| | | Dice | HD95 | Dice | HD95 | Dice | HD95 | Dice | HD95 |
| nnUnet(3D) | 17.8M | 85.61 | 4.33 | 78.72 | 6.59 | 70.23 | 4.91 | 78.19 | 5.28 |
| Trans-Unet(2D) | 96.1M | 84.42 | 4.91 | 75.12 | 7.18 | 73.97 | 5.07 | 77.84 | 5.72 |
| Swin-Unet(2D) | 79.6M | 90.64 | 6.01 | 80.78 | 7.07 | 75.54 | 5.98 | 82.32 | 6.35 |
| Swin-UNETR(3D) | 62.1M | 88.36 | 4.32 | 86.89 | 6.51 | 80.21 | 4.28 | 85.16 | 5.04 |
| AFTer-Unet(3D) | 41.5M | 91.93 | 4.15 | 87.15 | 6.76 | 81.58 | **3.91** | 86.89 | 4.94 |
| **MAT(3D)** | **11.7M** | **93.05** | **4.06** | **87.91** | **6.09** | **82.81** | 4.02 | **87.92** | **4.72** |

We compared our proposed 3D Medical Attention Transformer (MAT) model to other medical image segmentation models on the BraTS2018 dataset in Table 1. We selected several established models such as nnUnet (Isensee et al., 2021), Trans-Unet (Chen et al., 2021), Swin-Unet (Cao et al., 2021), Swin-UNETR (Hatamizadeh et al., 2022a), AFTer-Unet (Yan et al., 2022) for fair comparison. These models have shown state-of-the-art results in various medical image segmentation tasks and are widely used in the field.

Table 1 shows that all models have good segmentation performance in the Whole Tumor (WT) region, likely due to clear distinction between tumor and non-tumor regions and larger relative segmentation volume. For the Enhancing Tumor (ET) region, the most challenging to segment, 3D models incorporating the Transformer module achieved high Dice scores. The inclusion of the depth axis and long-range dependencies improves the model's understanding of the overall image, leading to better segmentation. 3D models outperformed 2D models in HD95 metrics due to problematic jagged edges when combining 2D slices into 3D images. MAT leverages 3D convolution to extract features and inputs feature maps with richer semantic information into the Axial Transformer module, achieving true global long-range dependency modeling, which resulting in superior performance. Overall, MAT achieved a mean Dice of 87.92% and a mean HD95 of 4.72 on the BraTS2018, surpassing AFTer-Unet in nearly all metrics.

In addition, the comparison of parameters demonstrates the superiority of our 3D axial attention algorithm as it reduces model complexity while preserving high segmentation performance. The MAT model also has a lower computational cost and can be trained on a single Tesla T4 (16GB) GPU, compared to other Transformer-based models.

## 3.3. Results on BraTS2021

Table 2 compares the performance of MAT with other medical image segmentation models on the BraTS2021 dataset. The comparison includes nnUnet (Isensee et al., 2021), Trans-

Table 2: Dice scores and HD95 of different methods on the BraTS2021 dataset (testing).

| Metrics | WT | | TC | | ET | | Mean | |
|---|---|---|---|---|---|---|---|---|
| | Dice | HD95 | Dice | HD95 | Dice | HD95 | Dice | HD95 |
| nnUnet(3D) | 92.14 | 7.33 | 89.56 | 3.94 | 83.67 | 4.02 | 88.46 | 5.10 |
| Trans-Unet(2D) | 91.73 | 7.92 | 85.47 | 6.02 | 81.25 | 4.68 | 86.15 | 6.21 |
| Swin-Unet(2D) | 93.51 | 7.51 | 90.64 | 5.61 | 85.34 | 4.18 | 89.83 | 5.77 |
| Swin-UNETR(3D) | **94.74** | 7.02 | 89.91 | 3.75 | **85.41** | 3.41 | 90.02 | 4.73 |
| AFTer-Unet(3D) | 93.47 | **6.95** | 90.48 | 3.76 | 85.31 | **3.29** | 89.75 | **4.67** |
| **MAT(3D)** | 93.21 | 7.13 | **91.91** | **3.56** | 85.05 | 3.61 | **90.06** | 4.77 |

Unet (Chen et al., 2021), Swin-Unet (Cao et al., 2021), Swin-UNETR (Hatamizadeh et al., 2022a), and AFTer-Unet (Yan et al., 2022). It is important to note that BraTS2021 contains more MRI data than BraTS2018, which may pose challenges for the lightweight MAT model.

Despite this, the results show that MAT's performance is comparable to other larger models. Notably, MAT performed significantly better than the other models in the TC (tumor core) region, with a Dice score improvement of 1.27% and a reduction of 0.20 in HD95 score. This highlights MAT's strong ability to learn with a limited number of parameters. Furthermore, on average, MAT is the best model under the Dice metric, and only slightly below the AFTer-Unet at 0.10 according to HD95.

## 3.4. Ablation study of warm-upped DLB

Table 3: Ablation study of DLB with warm-up schedule met hod on BraTS2018 and BraTS2021 Datasets through dice score.

| Mean Dice | BraTS 2018 | | BraTS 2021 | |
|---|---|---|---|---|
| | Training | Testing | Training | Testing |
| MAT without DLB | $85.97_{\pm 2.03}$ | $86.03_{\pm 2.52}$ | $89.51_{\pm 1.04}$ | $89.93_{\pm 1.52}$ |
| MAT with original DLB | $77.13_{\pm 2.49}$ | $79.22_{\pm 2.64}$ | $82.76_{\pm 1.49}$ | $82.92_{\pm 1.48}$ |
| **MAT with warm-upped DLB** | $\mathbf{87.65}_{\pm 1.85}$ | $\mathbf{87.92}_{\pm 2.01}$ | $\mathbf{90.02}_{\pm 0.98}$ | $\mathbf{90.06}_{\pm 1.47}$ |

Table 3 presents the results of ablation experiments to evaluate the efficacy of the proposed warm-upped DLB method on the BraTS2018 and BraTS2021 datasets. The parameters $\mathcal{T}$ and $\alpha$ were set to 3 and 1, respectively,as per the sets in (Shen et al., 2022). The results show improved performance of the MAT model with the warm-upped DLB method, with increases of 1.68% and 1.89% for the training and validation sets on the BraTS2018 dataset. The absence of warm-up led to decreased performance. The warm-upped DLB method was still effective on the BraTS2021 dataset, but with smaller improvement, indicating its greater efficacy for smaller datasets.

## 4. Conclusion

We propose 3D Medical Axial Transformer(MAT), a 3D end-to-end framework for brain tumor image segmentation that utilizes the axial Transformer and self-distillation scheme. The design of MAT enables efficient learning of semantic information while maintaining a lightweight architecture, making it suitable for clinical applications. Our experiments on brain tumor datasets demonstrate the superiority of MAT over previous related methods.

## Acknowledgments

We would like to thank everyone who supported us during this project, as well as the University of Tokyo for their financial support, which made it possible for us to complete this work. We also want to express our sincere appreciation to the hardworking healthcare professionals and AI architects in the field of medicine. Your dedication and contributions have played a crucial role in advancing healthcare. We strongly believe that a future will come when humans will no longer endure the hardships of illness and pain.

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

## Appendix A. Qualitative results on BraTS2018

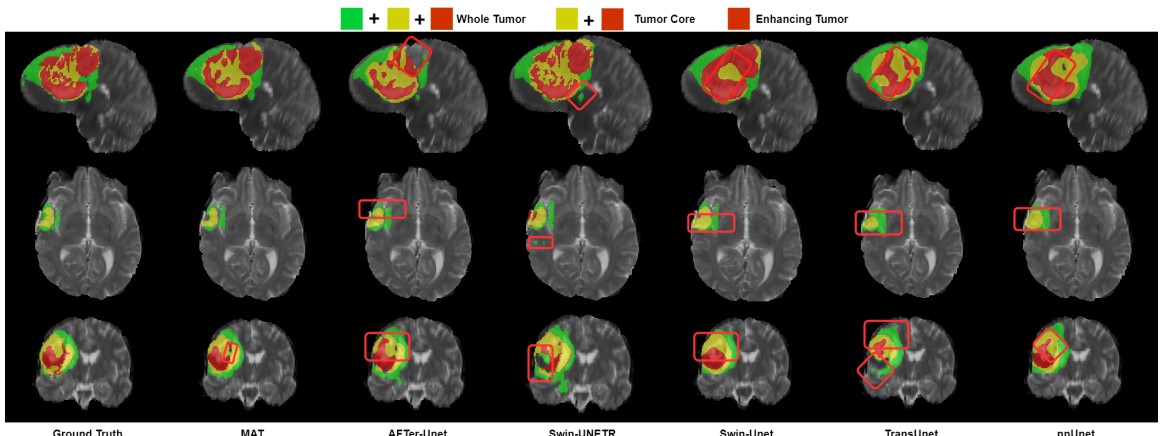

Figure 2: Qualitative results of different methods on the BraTS2018 dataset. The first row is sagittal, the second row is axial, and the third row is the coronal view. Red border is employed to mark the regions where segmentation obvious errors occur and to facilitate qualitative comparison of the model's results.

Figure 2 presents qualitative results of different models on the BraTS2018 dataset, compared with nnUnet (Isensee et al., 2021), Trans-Unet (Chen et al., 2021), Swin-Unet (Cao et al., 2021), Swin-UNETR (Hatamizadeh et al., 2022a), and AFTer-Unet (Yan et al., 2022).

We presented slices of images from three views (sagittal, axial, and coronal) to facilitate the comparison of 3D segmentation results. The most challenging regions to segment in brain tumor images are TC (yellow part + red part) and ET (red part) due to their small distinction from other tumors and small volume. The design of MAT's 3D Axial Transformer module with DLB enables it to learn global information, resulting in excellent segmentation ability for ET as seen in sagittal and coronal views. Additionally, as shown in the axial view (second row) and sagittal view (first row), MAT's predictions of gaps in tumors are more accurate due to its pixel-level long-distance dependence in three dimensions, in contrast to other models. Overall, MAT demonstrates a superior ability for multi-class tumor segmentation compared to other models, for both overall tumor region and local gaps.

# Appendix B. Additional Results on BraTS

Table 4: Mean and std of the Dice scores for various methods on the BraTS2021(testing).

| Dice | WT | TC | ET | Mean |
|---|---|---|---|---|
| nnUnet(3D) | $85.61_{\pm0.71}$ | $78.72_{\pm1.98}$ | $70.23_{\pm3.37}$ | $78.19_{\pm1.97}$ |
| Trans-Unet(2D) | $84.42_{\pm1.32}$ | $75.12_{\pm2.86}$ | $73.97_{\pm2.99}$ | $77.84_{\pm2.23}$ |
| Swin-Unet(2D) | $90.64_{\pm1.07}$ | $80.78_{\pm3.12}$ | $75.54_{\pm3.05}$ | $82.32_{\pm2.28}$ |
| Swin-UNETR(3D) | $88.36_{\pm0.89}$ | $86.89_{\pm2.75}$ | $80.21_{\pm2.84}$ | $85.16_{\pm1.92}$ |
| AFTer-Unet(3D) | $91.93_{\pm1.23}$ | $87.15_{\pm3.09}$ | $81.58_{\pm2.10}$ | $86.89_{\pm2.28}$ |
| **MAT(3D)** | $\mathbf{93.05}_{\pm1.49}$ | $\mathbf{87.91}_{\pm2.47}$ | $\mathbf{82.81}_{\pm2.41}$ | $\mathbf{87.92}_{\pm2.01}$ |

Table 5: Mean and std of the Dice scores for various methods on the BraTS2021(testing).

| HD95 | WT | TC | ET | Mean |
|---|---|---|---|---|
| nnUnet(3D) | $4.33_{\pm0.85}$ | $6.59_{\pm1.73}$ | $4.91_{\pm1.05}$ | $5.28_{\pm1.07}$ |
| Trans-Unet(2D) | $4.91_{\pm1.08}$ | $7.18_{\pm2.47}$ | $5.07_{\pm1.98}$ | $5.72_{\pm1.65}$ |
| Swin-Unet(2D) | $6.01_{\pm0.68}$ | $7.07_{\pm2.56}$ | $5.98_{\pm2.48}$ | $6.35_{\pm1.63}$ |
| Swin-UNETR(3D) | $4.32_{\pm0.71}$ | $6.51_{\pm1.91}$ | $4.28_{\pm1.87}$ | $5.04_{\pm1.40}$ |
| AFTer-Unet(3D) | $4.15_{\pm0.88}$ | $6.76_{\pm2.39}$ | $\mathbf{3.91}_{\pm2.04}$ | $4.94_{\pm1.61}$ |
| **MAT(3D)** | $\mathbf{4.06}_{\pm1.01}$ | $\mathbf{6.09}_{\pm2.16}$ | $4.02_{\pm2.10}$ | $\mathbf{4.72}_{\pm1.64}$ |

Table 6: Mean and std of the Dice scores for various methods on the BraTS2021(testing).

| Dice | WT | TC | ET | Mean |
|---|---|---|---|---|
| nnUnet(3D) | $92.14_{\pm0.69}$ | $89.56_{\pm1.83}$ | $83.67_{\pm2.39}$ | $88.46_{\pm1.41}$ |
| Trans-Unet(2D) | $91.73_{\pm1.01}$ | $85.47_{\pm1.37}$ | $81.25_{\pm2.19}$ | $86.15_{\pm1.40}$ |
| Swin-Unet(2D) | $93.51_{\pm0.87}$ | $90.64_{\pm1.97}$ | $85.34_{\pm2.12}$ | $89.83_{\pm1.53}$ |
| Swin-UNETR(3D) | $\mathbf{94.74}_{\pm0.78}$ | $89.91_{\pm1.81}$ | $\mathbf{85.41}_{\pm1.97}$ | $90.02_{\pm1.39}$ |
| AFTer-Unet(3D) | $93.47_{\pm0.93}$ | $90.48_{\pm2.01}$ | $85.31_{\pm2.14}$ | $89.75_{\pm1.58}$ |
| **MAT(3D)** | $93.21_{\pm0.93}$ | $91.91_{\pm1.92}$ | $85.05_{\pm1.98}$ | $90.06_{\pm1.47}$ |

Table 7: Mean and std of the Dice scores for various methods on the BraTS2021(testing)

| HD95 | WT | TC | ET | Mean |
|---|---|---|---|---|
| nnUnet(3D) | $7.33_{\pm1.37}$ | $3.94_{\pm1.08}$ | $4.02_{\pm1.32}$ | $5.10_{\pm1.11}$ |
| Trans-Unet(2D) | $7.92_{\pm1.22}$ | $6.02_{\pm1.30}$ | $4.68_{\pm1.47}$ | $6.21_{\pm1.20}$ |
| Swin-Unet(2D) | $7.51_{\pm1.02}$ | $5.61_{\pm1.19}$ | $4.18_{\pm1.28}$ | $5.77_{\pm1.10}$ |
| Swin-UNETR(3D) | $7.02_{\pm0.96}$ | $3.75_{\pm0.99}$ | $3.41_{\pm1.13}$ | $4.73_{\pm0.92}$ |
| AFTer-Unet(3D) | $\mathbf{6.95}_{\pm1.00}$ | $3.76_{\pm1.34}$ | $\mathbf{3.29}_{\pm0.98}$ | $\mathbf{4.67}_{\pm1.02}$ |
| **MAT(3D)** | $7.13_{\pm1.10}$ | $3.56_{\pm1.38}$ | $3.61_{\pm1.29}$ | $4.77_{\pm1.08}$ |

## Appendix C. Formulas of evaluation metrics

$$\text{Dice}(T, P) = \frac{2\sum_{i=1}^{I} T_i P_i}{\sum_{i=1}^{I} T_i + \sum_{i=1}^{I} P_i} \tag{5}$$

$$HD\left(T', P'\right) = \max\left\{\max_{t' \in T'} \min_{p' \in P'} \left\|t' - p'\right\|, \max_{p' \in P'} \min_{t' \in T'} \left\|p' - t'\right\|\right\} \tag{6}$$

Where $T_i$ and $P_i$ denote the ground truth and predicted values of voxel $i$, while $T'$ and $P'$ denote the set of surface points of the ground truth and predicted values, respectively. HD95 is based on the calculation of the 95th percentile of the distances between boundary points in T' and P', in order to eliminate the effect of a very small subset of the outliers.

## Appendix D. The output of axial attention for depth axis and width axis

$$y_{\text{depth}_{ijk}} = \sum_{d=1}^{D} \text{Softmax}\left(q_{ijk}^{T} k_{djk} + q_{(i,j,k)}^{T} r_{djk}^{q} + k_{djk}^{T} r_{djk}^{k}\right)\left(b_{djk} + r_{djk}^{v}\right) \tag{7}$$

$$y_{\text{width}_{ijk}} = \sum_{w=1}^{W} \text{Softmax}\left(q_{ijk}^{T} k_{ijw} + q_{(i,j,k)}^{T} r_{ijw}^{q} + k_{ijw}^{T} r_{ijw}^{k}\right)\left(v_{ijw} + r_{ijw}^{v}\right) \tag{8}$$

