# OpenReview forum: "3D Medical Axial Transformer: A Lightweight Transformer Model for 3D Brain Tumor Segmentation"
_MIDL.io/2023/Conference — MIDL 2023 Poster_

### Official Review · Reviewer_s4Hv · 2023-02-01

**Confidence:** 4
**Preliminary Rating:** 4

**Summary:**

a lightweight 3D transformer was proposed for medical image segmentation. The authors mainly use axial transformer to reduce the model size, and combine it with a U-Net like structure. Results on BraTS dataset showed its effectiveness. It was compared with an existing 3D transformer. The authors also employed self-distillation of the model for performance improvement.

**Strengths:**

1, The motivation of developing lightweight models is clear, and the proposed model can reduce the computational cost for medical image segmentation.

2, The authors conducted experiments on two public datasets to show its effectiveness.

3, The paper is generally smooth and easy to follow.


**Weaknesses:**

1, The novelty of the proposed method is a bit limited. The main idea in this work is axial transformer, but it has been previously proposed by Valanarasu et al., 2021). The self-distillation method has also been previously proposed by Shen et al. 2022.

2, In the experiment, the authors only compared the proposed model with one 3D transformer network (AFTerUNet). Recently, there have been several works on 3D transformer such as CoTR mentioned by the authors. The proposed method should be compared with these models.

3, The authors claimed to propose a lightweight model, but the model size was not listed in the text. It would be good to provide such information in Table I.


**Deanonymize Review:**

no

**Detailed Comments:**

1, The Dice and HD values were only given with mean values. How about the std? In addition, statistical testing results are encouraged to be given to show if the performance improvement was significant.

2, The implementation of self-distillation was not clear. Shen et al keep half of the mini-batch the same in two neighboring forward passes, did the authors implement in the same way? Also, there was no ablation study to show the effectiveness of the self-distillation.


**Paper Type:**

methodological development

**Questions To Address In The Rebuttal:**

1, My main concern is the novelty of this paper as mentioned above.

2, It would be more convincing if the authors compare the proposed method with more existing 3D transformers for medical image segmentation, for example, CoTR and Swin UNETR.

---

### Official Review · Reviewer_pjwy · 2023-02-03

**Confidence:** 4
**Preliminary Rating:** 2
**Recommendation:** Poster

**Summary:**

The paper proposes a lightweight transformer model for 3D medical image segmentation. The model consists of transformer encoder and CNN decoder. The method incorporates axial attention, which applies attention to each dimension separately, and self-distillation to achieve remarkable performance on the Brats18 and Brats21 datasets.

**Strengths:**

- relevant selection of the baselines
- the method description is clear, easy to follow, and the architecture seems to be easy to implement
- the transformer-based method that can directly utilize 3d images

**Weaknesses:**

- poor organization of the results section. Large images from Qualitative results could be moved to the appendix or compressed. It's confusing that validation on the larger dataset is missing in the main text. it would be helpful to have data from appendix A in the results table, as well as FLOPs count and computational time, as transformers typically are more computationally expensive.
Data from Appendix C would be also helpful in Tables 1 and 2.

- design of the experiments
In section 3.1.1, were hyperparameters tuned using the validation split, and how is the final model selected? CV scheme or testing dataset is missing in the experiment design.


minor:
Evaluation metrics: formulas are redundant and can be moved to appendix if needed. The freed space can be used for brats 2021 results
What does "HD95 is the output of HD multiplied by 95%" mean?

**Deanonymize Review:**

no

**Paper Type:**

methodological development

**Questions To Address In The Rebuttal:**

Could you please address the 2 points from the weaknesses section, namely rework the results section including additional data, and clarify the design of the experiment, the hyperparameter optimization, and the process of the best model selection.

---

### Official Review · Reviewer_WVnR · 2023-02-06

**Confidence:** 4
**Preliminary Rating:** 4
**Recommendation:** Poster

**Summary:**

The author developed a lightweight 3D segmentation transformer based upon the axial attention. A self-distillation method is involved during the optimization of the network. The evaluation is conducted on the BraTS dataset and the results on BraTS2018 is well improved comparing with other methods. The 'lightweight' property is shown by the number of parameters and the effectiveness of the self-distillation is proved by an ablation study.

**Strengths:**

- The proposed method is clearly explained so the idea is very easy to follow.
- The experiment result looks promising and the stats of the model parameters and the ablation study illustrate the effectiveness of the axial attention mechanism and the self-distillation.

**Weaknesses:**

(1) There is lack of novelty in terms of methodology since the paper is a direct implementation work that combines:
>- The axial attention idea from (Valanarasu et al, 2021)
>- The self-distillation idea from (Shen et al, 2022)

(2) In the evaluation, the UNETR should be included as a baseline method as it is one the most popular application of 3D transformer in medical image segmentation.

(3) Some of the annotation in the paper need to be modified to avoid potential confusion.
>- In the equation for the axial attention, the query, key and value vectors should be lowercase. The current version is confusing as they are the same with the superscripts. The subscript indicating the position should be consistent.
>- In 2.2.2, avoid using T1, T,2 T3 to annotate the stages as T1 and T2 are modalities as well.


**Deanonymize Review:**

no

**Detailed Comments:**

(1) In the evaluation, the UNETR should be included as a baseline method as it is one the most popular application of 3D transformer in medical image segmentation.

(2) Some of the annotation in the paper need to be modified to avoid potential confusion.
>- In the equation for the axial attention, the query, key and value vectors should be lowercase. The current version is confusing as they are the same with the superscripts. The subscript indicating the position should be consistent.
>- In 2.2.2, avoid using T1, T,2 T3 to annotate the stages as T1 and T2 are modalities as well.

**Paper Type:**

validation/application paper

**Questions To Address In The Rebuttal:**

(1) The author claim that the self-distillation is effective for small datasets, and the result seems to be reasonable (BraTS2018 is a small dataset and the proposed model do better)
>- What if you use this on AFTer-Unet? Will the result Dice surpass 87.92? Or in other words, is it necessary to consider the whole axis for local targets like tumor?
>- Intuitively, the self-distillation seems to be slowing down the optimization as it use the result from the previous epoch as a contraint. Is it true in your experiment?

(2) Will the code be available?

---

### Official Review · Reviewer_eQG6 · 2023-02-06

**Confidence:** 4
**Preliminary Rating:** 4
**Recommendation:** Poster

**Summary:**

This work proposed a lightweight transformer model (MAT) in integrating 1) gated axial-attention blocks and 2) warm-upped self-distillation from last mini-batch (DLB) method. The proposed method was evaluated on both BraTS2018 and BraTS2021 datasets. The experimental results showed that MAT outperformed 2 2D and 2 3D SOTAs with fewer parameters, especially on the BraTS2018 dataset.

**Strengths:**

1. The manuscript is overall well-written.
2. The proposed MAT achieved better or similar performance compared with 4 SOTA methods on BraTS2018 and BraTS2021 datasets.
3. An ablation study has been conducted to demonstrate the importance of warm-upped in DLB, especially for small datasets.

**Weaknesses:**

1. The novelty of MAT is somewhat limited since the proposed structure is fundamentally the same as the local branch of MedT (Valanarasu et al., 2021).
2. The generalizability of MAT may be limited since it is only evaluated on brain tumor segmentation (BraTS datasets) and the improvement becomes less significant with more training data.

**Deanonymize Review:**

no

**Detailed Comments:**

1. Please provide mean and std in Table 1, Table 2, Table 3, and Figure 3.

**Paper Type:**

methodological development

**Questions To Address In The Rebuttal:**

1. Please elaborate on the fundamental difference between MAT and MedT (Valanarasu et al., 2021).
2. Figure 2 is not well explained.
2.a What do the red boxes represent exactly?
2.b What do the blue dash lines mean?
2.c Additionally, it would be better to demonstrate cases with the best, medium, and worst performances.
2.d In the legend, both green+yellow+red and yellow+red are labeled as "whole tumor". Is it a typo?

---

### Meta-Review · Area_Chair_AMpS · 2023-02-18

**Recommendation:** Accept (Poster)
**Confidence:** 4

**Metareview:**

The reviewers appreciated the experimental and writing quality of this paper, which introduces a lightweight Transformer model. A majority of the reviewers expressed their recommendation of “weak accept”. The rebuttal for the negative review (“weak reject”) is adequate. Thus, my recommendation leans towards acceptance.